# Dilated-Left Ventricular Non-Compaction Cardiomyopathy in a Pediatric Case with *SPEG* Compound Heterozygous Variants

**DOI:** 10.3390/ijms23095205

**Published:** 2022-05-06

**Authors:** Hager Jaouadi, Fedoua El Louali, Chloé Wanert, Aline Cano, Caroline Ovaert, Stéphane Zaffran

**Affiliations:** 1Aix Marseille University, INSERM, Marseille Medical Genetics, U1251 Marseille, France; hajer.jaouadi@univ-amu.fr (H.J.); chloe.wanert@ap-hm.fr (C.W.); caroline.ovaert@ap-hm.fr (C.O.); 2Department of Paediatric and Congenital Cardiology, Timone Hospital Marseille, University Hospital, 13005 Marseille, France; fedoua.el-louali@ap-hm.fr; 3Reference Center of Inherited Metabolic Disorders, Timone Hospital Marseille, University Hospital, 13005 Marseille, France; aline.cano@ap-hm.fr

**Keywords:** cardiac magnetic resonance imaging (MRI), dilated-LVNC, de novo variant, echocardiography, in silico analysis tools, *SPEG* gene

## Abstract

Left Ventricular Non-Compaction (LVNC) is defined by the triad prominent myocardial trabecular meshwork, thin compacted layer, and deep intertrabecular recesses. LVNC associated with dilation is characterized by the coexistence of left ventricular dilation and systolic dysfunction. Pediatric cases with dilated-LVNC have worse outcomes than those with isolated dilated cardiomyopathy and adult patients. Herein, we report a clinical and genetic investigation using trio-based whole-exome sequencing of a pediatric case with early-onset dilated-LVNC. Compound heterozygous mutations were identified in the Striated Muscle Enriched Protein Kinase (*SPEG*) gene, a key regulator of cardiac calcium homeostasis. A paternally inherited mutation: *SPEG*; p.(Arg2470Ser) and the second variant, *SPEG*; p.(Pro2687Thr), is common and occurred de novo. Subsequently, Sanger sequencing was performed for the family in order to segregate the variants. Thus, the index case, his father, and both sisters carried the *SPEG*: p.(Arg2470Ser) variant. Only the index patient carried both *SPEG* variants. Both sisters, as well as the patient’s father, showed LVNC without cardiac dysfunction. The unaffected mother did not harbor any of the variants. The in silico analysis of the identified variants (rare and common) showed a decrease in protein stability with alterations of the physical properties as well as high conservation scores for the mutated residues. Interestingly, using the Project HOPE tool, the *SPEG*; p.(Pro2687Thr) variant is predicted to disturb the second fibronectin type III domain of the protein and may abolish its function. To our knowledge, the present case is the first description of compound heterozygous *SPEG* mutations involving a de novo variant and causing dilated-LVNC without neuropathy or centronuclear myopathy.

## 1. Introduction

Left Ventricular Non-Compaction (LVNC) is defined by the triad: prominent myocardial trabecular meshwork, thin compacted layer, and deep intertrabecular recesses [1,2]. It is characterized by incomplete penetrance and variable expressivity. Of note, the anatomy of the trabeculae is different among individuals and can be observed in healthy subjects [3]. Thus, the characterization of pathological LVNC remains challenging.

The clinical diagnosis of LVNC is based mostly on imaging by transthoracic echocardiography. According to diagnostic criteria by Petersen et al., LVNC can be diagnosed if the ratio of non-compacted to compacted myocardium is >2.3 at end-diastole [4]. However, a consensual standardization is lacking for LVNC diagnosis criteria, especially in children.

Although LVNC mainly affects the left ventricle, the right ventricle can be affected as well. Moreover, cases with LVNC in conjunction with a wide range of cardiac disorders have been observed. Therefore, several subtypes of LVNC have been described including dilated-LVNC, hypertrophic-LVNC, mixed dilated-hypertrophic LVNC phenotype, and isolated LVNC [2].

In the pediatric population, LVNC accounts for nearly 10% of all primary cardiomyopathies [5,6]. Moreover, pediatric cases with dilated-LVNC had the worst outcomes followed by those with hypertrophic-LVNC [7,8]. The clinical manifestation in patients with LVNC such as heart failure, arrhythmias, and embolic events are comparable in the adult and pediatric populations [8,9].

The genetic diagnostic yield of LVNC is overall low and highly variable, ranging from 9 to 41% depending on the size of the patient cohort and gene testing strategy (gene panels, WES…) [1,10,11]. Moreover, owing to the small cohort sizes, little is known about LVNC genotype-phenotype correlations.

Emerging data reporting genetic causes of LVNC support an overlap in the etiological basis between this entity and the other types of cardiomyopathies [2,3].

In the present study, we report clinical and genetic investigation of a pediatric case with early-onset dilated-LVNC and asymptomatic family members with the LVNC phenotype. Our study revealed two missense variants in the *SPEG* gene with a high prediction of pathogenicity, which when acting in concert seem to cause a severe phenotype of the disease, as observed here.

## 2. Results

### 2.1. Clinical Presentation and WES Findings

The index patient, a 7-year-old boy (weight = 26.6 Kg (+0.5 SD); height = 125 cm (+0.5 SD)), was born at full term to ostensibly healthy, non-consanguineous parents. Family history was normal with no sudden death nor miscarriages. The neonatal period was uneventful without feeding or growth difficulties. At the age of six months, he was referred to the pediatric cardiology department for unexplained tachypnea. Cardiac ultrasound revealed dilated-LVNC (LV end-diastolic z-value +2.2) with significantly altered LV systolic function (ejection fraction estimated at 32% (Simpson analysis) and non-compaction of the LV. The right ventricle was unaffected. Cardiac MRI performed at the age of 8 months confirmed the diagnosis (Figure 1a. He was started on diuretics, ACE inhibitors, and oral anti-aggregation therapy (aspirin). His clinical course was favorable, with progressive clinical and echocardiographic improvement. The latest echocardiography, performed at the age of 7 years (Figure 1b), showed a non-dilated LV with isolated LVNC and subnormal systolic function (EF 50–55%). The patient remained solely on ACE inhibitors. Of note, the patient presented mild hyperlordosis with fluctuating CPK levels; however, his muscular biopsy was normal without objective clinical muscle deficit. No developmental delay was noted. Ophthalmic examination was normal.

Whole-exome sequencing analysis allowed us to prioritize two missense heterozygous variants in the *SPEG* gene, one which was a paternally inherited variant: NM_005876: exon30: c.7408C>A; p.(Arg2470Ser) (MAF = 8.422e^−5^ in the gnomAD database). This variant is located between the first protein kinase domain and the Ig-like 9 domain of the gene. The second variant, NM_005876: exon34: c.8059C>A; p.(Pro2687Thr) (MAF = 0.12), occurred de novo. This variant is located in the second fibronectin type-III domain of the gene (Figure 2A). Subsequently, Sanger sequencing was performed for the family members in order to segregate the variants. Thus, the index-case (II-2), his father (I-1), and both sisters (II-1; II-3) carried the *SPEG*: p.(Arg2470Ser) variant. Only the index patient carried both *SPEG* variants. The unaffected mother did not harbor any of the variants (Figure 2B).

Of note, no other clinically relevant variants that may explain the phenotype in this family were found.

Family echocardiographic screening detected LVNC with preserved LV function in the index patient’s father (44 years old) and his older sister (10 years old). The diagnosis was confirmed in both with cardiac MRI (Figure 1-3,4). Both patients were asymptomatic.

His younger sister, 4 years old, was shown to have a trabeculated left ventricle but without criteria of non-compaction and with preserved systolic function (Figure 1-5). MRI assessment has been scheduled at an older age. No other clinical abnormalities were observed in those three first-degree relatives. The mother’s echocardiogram was completely normal.

### 2.2. In Silico Analysis

Both variants are predicted deleterious by at least three in silico prediction tools. More specifically, the *SPEG*; p.(Arg2470Ser) variant has been predicted deleterious by Mutation taster, SIFT, and UMD predictor, and the p.(Pro2687Thr) variant has been predicted deleterious by Provean, Polyphen2_HDIV, and _HVAR. Interestingly, the common *SPEG*; p.(Pro2687Thr) variant had a higher Combined Annotation Dependent Depletion Phred score (score v1.3 = 27) compared to the rare *SPEG*; p.(Arg2470Ser) variant (score v1.3 = 25.8). The conservation analysis by ConSurf (https://consurf.tau.ac.il/; accessed date 10 December 2021) showed high conservation scores for the *SPEG*; Arg2470 and Pro2687 residues of 9 and 8, respectively.

The I-Mutant 2.0 software was used to predict protein stability [12]. Both variants showed decreased protein stability with a reliability index of 6 for the *SPEG*; p.(Arg2470Ser) variant and 9 for the *SPEG*; p.(Pro2687Thr) variant. Protein stability changes were calculated at pH = 7 and 25 °C.

To assess the physical properties and interactions of the protein, the Project HOPE tool was acquired [8]. The results of this analysis for the *SPEG*; p.(Arg2470Ser) rare variant showed that the mutant residue (Serine) is less smaller than the wild-type residue (Arginine). Moreover, the charge of the Arginine residue at position 2470 was positive, and the mutant residue charge was neutral. Thus, the mutant residue is more hydrophobic than the wild-type residue, which can result in loss of hydrogen bonds and/or disturb correct folding. Similarly, the *SPEG*; p.(Pro2687Thr) variant, located in the fibronectin III domain, resulted in a substitution with a residue (Threonine) with a different size and different hydrophobicity properties. The wild-type residue (Proline) is more hydrophobic than the mutant residue. Project HOPE predicted that the p.(Pro2687Thr) variant can disturb this domain and abolish its function. Furthermore, Prolines are known to be very rigid and therefore induce a special backbone conformation which might be required at this position. The variant can disturb this special conformation.

Considering the segregation of the variants with the disease in the family and the favor pathogenic prediction, the *SPEG*; p.(Pro2687Thr) de novo variant seems to be a key contributor to the early onset and the severity of the disease in the presence of the rare inherited variant *SPEG*; p.(Arg2470Ser).

## 3. Discussion

*SPEG* (Striated Muscle Enriched Protein Kinase) is a protein-coding gene. The protein belongs to the myosin light chain kinase family, which is required for myocyte cytoskeletal development [13]. Due to extensive alternative splicing, the *SPEG* complex locus encodes several isoforms, including: Aortic Preferentially Expressed Protein 1 (Apeg), Brain Preferentially Expressed Protein (Bpeg), Speg-α, and Speg-β. Contrary to Apeg1 and Bpeg, Speg-α and Speg-β contain two kinase domains and are predominantly expressed in skeletal and cardiac muscle [13]. In vivo studies in mice reported a drastic shift in isoform predominance in the heart from Apeg1 during the embryonic stage to Speg-α and -β in the postnatal period, which can be correlated to the cardiomyocyte maturation occurring in this period [13]. Of note, Liu X and colleagues reported the presence of the four isoforms in fetal hearts [14]. In the same study, the authors demonstrated the implication of *SPEG* locus in dilated cardiomyopathy. In fact, homozygous *SPEG* mutant mice led to dilation of right and left atria and ventricles with increased neonatal mortality [14].

Interestingly, along with the Desmin gene (*DES*), the expression of *SPEG* gene may be controlled by the desmin locus control region, and both genes shared similar tissues expression profiles [15]. Of note, inherited and de novo mutations in the *DES* gene have been linked to myofibrillar myopathy and dilated and arrhythmogenic cardiomyopathies [16,17,18,19]. More recently, a de novo missense mutation in the *DES* gene has been associated to LVNC cardiomyopathy [20].

Firstly, mutations in the *SPEG* gene have been identified in patients with centronuclear myopathy type 5 (phenotype MIM number 615959) with an autosomal recessive inheritance pattern [21]. Centronuclear myopathies are a group of rare inherited neuromuscular disorders characterized by a progressive weakness defined mainly by numerous centrally placed nuclei on a muscle biopsy and a highly variable clinical presentation including ophthalmoplegia, respiratory failure, and scoliosis [21,22]. Strikingly, Lornage et al. reported a patient affected by myopathy without central nuclei [23].

In 2014, Agrawal et al. identified *SPEG* mutations in three patients with centronuclear myopathy, two of whom presented dilated cardiomyopathy [21]. Thereafter, patients with non-syndromic dilated cardiomyopathy and no myopathy features or skeletal muscle involvement were reported [24,25]. Although several cases of dilated cardiomyopathy linked to *SPEG* mutations were reported either associated to myopathy or not, only one case of non-compaction and neuropathy was reported [26]. His cardiac evaluation showed a disclosed enlarged atria, abnormal trabeculation of the left ventricle, and intratrabeculation recesses [26]. Taking together, the *SPEG* gene seems to be a pleiotropic gene giving rise to different diseases and cardiac phenotypes with variable expressivity.

Recently, different studies showed the role of *SPEG* in calcium reuptake into the sarcoendoplasmic reticulum through regulating SERCA2a (Sarcoplasmic/Endoplasmic Reticulum Calcium ATPase2) in cardiomyocytes, which is a crucial process in the excitation-contraction coupling [27,28]. Genes implicated in Ca^2+^ handling such as *TTN*, *RYR2*, *CASQ2*, and *RBM20* have been reported in patients with LVNC [29,30]. Similarly, the *SPEG* gene was defined as a key regulator of cardiac calcium handling owing to its role in Ca^2+^ regulation through maintenance of transverse tubule formation and phosphorylation of junctional membrane complex formation [27,28]. Inducible deletion of *SPEG* inhibited SERCA2a calcium transporting activity and impaired Ca^2+^ reuptake into the sarcoplasmic reticulum through phosphorylation at the Thr484 SERCA2a phosphorylation site [27]. Aberrant calcium handling leads mainly to altered contractility and arrhythmogenesis. Furthermore, Quick et al. showed that the Speg protein binds to both Jph2 (Junctophilin2) and RyR2 proteins. Given that Speg binds to both RyR*2* and SERCA2a, it plays a role in the structural organization of sarcoplasmic reticulum complexes which may explain its implication in cardiomyopathies [31].

To date, 15 mutations in the *SPEG* gene have been linked to cardiac phenotypes in humans through either an autosomal recessive pattern of inheritance or a heterozygous compound model [21,22,23,24,25,26,32,33]. Furthermore, the majority of protein-truncating variants have been found in patients with myopathy associated with dilated cardiomyopathy (Table 1). However, a clear-cut correlation between the phenotype and the variant type, location, or genotype is lacking. Further studies involving a large cohort of patients harboring *SPEG* mutations are necessary to evaluate a potential genotype–phenotype correlation.

In the present study, we report an early-onset dilated-LVNC case without neuropathy or myopathy features. Furthermore, the cardiac evaluation of both sisters and the father carrying the rare *SPEG* variant p.(Arg2470Ser) showed an LVNC phenotype without an altered cardiac function. These findings support the fact that the common *SPEG*; p.(Pro2687Thr) variant is functionally relevant and may act as a risk allele in the presence of the rare *SPEG* variant.

Further characterization of *SPEG* variants is needed to better understand their physiopathological mechanism and cumulative effect.

## 4. Materials and Methods

### 4.1. Whole-Exome Sequencing (WES)

Peripheral blood sample was collected after obtaining the written informed consent. Genomic DNA was extracted by standard techniques.

Whole-exome sequencing was performed for the patient and his parents using the Agilent SureSelect Human All Exon kit V4 (Agilent Technologies, Santa Clara, CA, USA). The captured libraries were sequenced on Illumina NextSeq500 sequencing platform (Illumina, San Diego, CA, USA). Raw fastQ files were aligned to the hg19 reference human genome (University of California Santa Cruz, UCSC) using BWA software. Variant calling workflow was performed according to the GATK best practices. The output files were annotated using ANNOVAR software.

### 4.2. In Silico Analysis Tools

#### 4.2.1. Variant Prioritization

Variant prioritization was carried out using Variant Annotation and Filtering Tool (VarAFT http://varaft.eu/) version 2.17; accessed date 10 December 2021. To pinpoint candidate causative variants, we adopted the following filtering strategy: firstly, a minimum variant allele frequency of 2% was applied using gnomAD database (http://gnomad.broadinstitute.org/; accessed date 10 December 2021). Then, we removed non-coding and synonymous variants. The remaining variants including missense and protein-truncating variants were filtered based on their in silico pathogenicity prediction to assess their likely causality.

A pedigree-based analysis was performed considering autosomal dominance and recessiveness as well as X-linked dominance and recessiveness patterns of inheritance.

#### 4.2.2. Residue Conservation Analysis

Residue conservation was determined using the ConSurf server (https://consurf.tau.ac.il/; accessed date: 4 January 2022). ConSurf is a bioinformatics tool for estimating the evolutionary conservation of amino and nucleic acid positions in a protein/DNA/RNA molecule based on the phylogenetic relations between homologous sequences. The degree to which an amino or nucleic acid position is evolutionarily conserved is strongly dependent on its structural and functional importance [34].

#### 4.2.3. Protein Physical Properties, Stability, and Interaction Assessment

The prediction of variants’ effects on the protein was performed using Project HOPE and I-Mutant 2.0 software [12,35].

A flowchart explaining the study methodology is summarized in Figure 3.

## Figures and Tables

**Figure 1 ijms-23-05205-f001:**
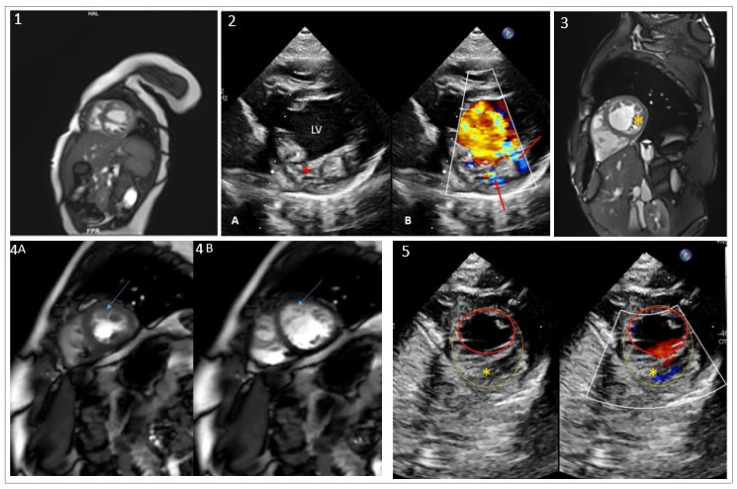
Echocardiograms, MRI images of the index patient and his relatives, family pedigree and variant conservation score. (**1**) MRI image of the index patient at the age of 8 months shows dilated-LVNC with altered LV function. (**2**) Transthoracic echocardiography of the index patient: (**2A**) Parasternal short axis view showing the left ventricle (LV) with increased trabeculations (red asterisk). (**2B**) Color Doppler analysis, the red arrows show the blood flow in the intratrabecular recesses. (**3**) MRI image of the index patient’s father showing LVNC (yellow asterisk) (subject I-1). (**4**) MRI images of subject II-1 showing LV short axis in systolic (**4A**) and diastolic (**4B**) phases with LVNC (blue arrow). (**5**) Echocardiography of subject II-3 showing trabeculated LV (yellow asterisk). An endocardial border (red circle) and non-compacted layer border to include the trabeculated area (yellow circle) are drawn.

**Figure 2 ijms-23-05205-f002:**
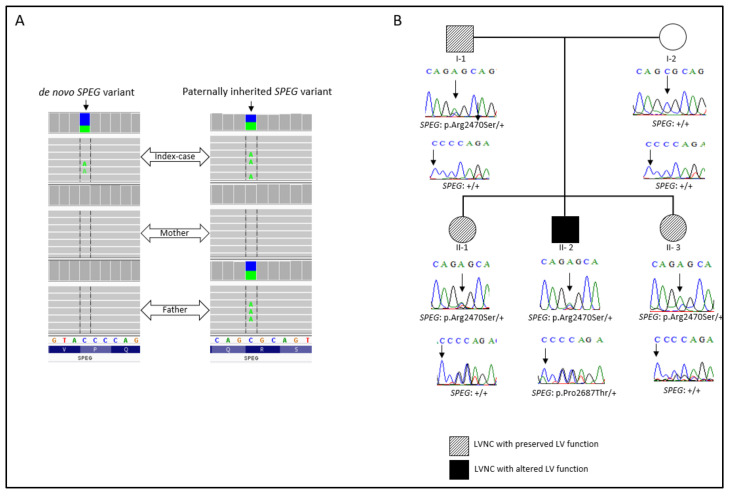
(**A**) Integrative genomics viewer visualization showing the de novo and the inherited *SPEG* variants. (**B**) Family pedigree. Sanger electropherograms and genotypes are shown below symbols: (+) indicates the wild-type allele, and the arrow indicates the position of the mutation.

**Figure 3 ijms-23-05205-f003:**
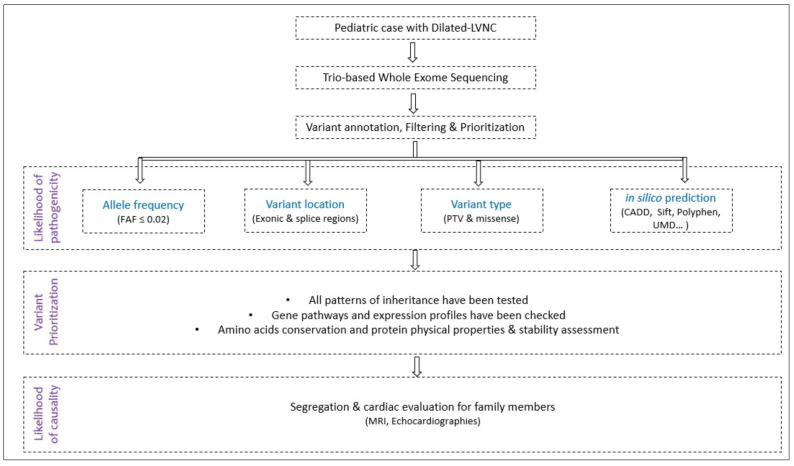
Study design flowchart. (FAF: Filtering Allele Frequency; PTV: Protein-Truncating Variants).

**Table 1 ijms-23-05205-t001:** *SPEG* mutations reported in patients with cardiac phenotypes.

Gene	cDNA_Change	Protein_Change	Cardiac Phenotype	Myopathy	Reference
*SPEG*	c.1071_1074dup; c.4399C>T	Lys359Valfs*35; Arg1467*	Reduced myocardial contraction	Present	[23]
*SPEG*	c.2183delT; c.8962_8963insCGGG GCGAACGTTCGTG GCCAAGAT	Leu728Argfs*82; Val2997Glyfs*52	Sinus tachycardia	Present	[32]
*SPEG*	c.2915_2916delCCinsA; c.8270G>T	Ala972Aspfs*79; Gly2757Val	DCM; Mitral insufficiency	Present	[21]
*SPEG*	c.3709_3715 + 29del36; c.4276C>T	Thr1237Serfs*46; Arg1426*	DCM	Present	[21]
*SPEG*	c. 5038G>A	Glu1680Lys	DCM	Absent	[25]
*SPEG*	c.6697C>T	Gln2233*	NA	Present	[21]
*SPEG*	c.7119C>A	Tyr2373*	LVNC/enlarged atria	Present	[26]
*SPEG*	c.7408C>A; c.8059C>A	Arg2470Ser; Pro2687Thr	DCM-LVNC	Absent	Present study
*SPEG*	c.8710A>G	Thr2904Ala	DCM	Present	[22]
*SPEG*	c.9028_9030delGAG	Glu3010del	DCM	Absent	[24]
*SPEG*	c.9185_9187delTGG	Val3062del	DCM/Severe mitral valve insufficiency	Present	[32]
*SPEG*	c.9586C>T	Arg3196*	Fetal bradycardia, DCM, Mild mitral insufficiency	Mild	[33]

## Data Availability

The authors confirm that the data supporting the findings of this study are available within the article.

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
