# Peer review of "Dilated-Left Ventricular Non-Compaction Cardiomyopathy in a Pediatric Case with SPEG Compound Heterozygous Variants"

_ijms, 2022, doi:10.3390/ijms23095205_

Round 1

Reviewer 1 Report

In this paper entitled “Expanding the Clinical and Genetic Spectrum of SPEG Variants: A Pediatric Case of Dilated-Left Ventricular Non-Compaction Cardiomyopathy with Compound Heterozygous Variants”, Jaouadi et al. describe the causative role of two compound mutations in the SPEG gene associated with Left Ventricle Non Compaction (LVNC) in a 7-year-old-boy.

The LVNC is quite a challenging phenotype and this is clear to the authors too as well as the underlying genetic etiology. Under this light the contribution of the authors in describing novel genes or novel inheritance patterns (as in this case autosomal recessive) is of paramount importance. However, I have the following points to submit to the authors:

  1. Since the index case is a boy (7-year-old) I would consider implementing the Introduction section mentioning how the LVNC behaves among the pediatric population (incidence, ect);
  2. It is not required anymore at the previsional level to mention or allocate dedicated figures to what is happening at the amino acid level when a non-synonymous mutation occur (i.e. Figure 3 and 4 as well as the dedicated Methods sections): all of the software’s doing NGS analysis take into account this information. However if the authors would like to have such data, I can suggest using PyMol or crystal structure modelling soft that have better impact on the reader
  3. In the summary there are bad writing at the genetic level nomenclature (i.e. SPEG; p. P2687T instead of SPEG:p.P2687T and so on…)
  4. It would be better maybe to add a table stating the presently genes associated with LVNC, the inheritance pattern (AD, AR, X-linked..), phneotypes (LVNC isolated or associated with HCM, DCM, FA, Syndromic features etc)

Reviewer 2 Report

Hager Jaouadi and colleagues submitted a research article entitled: Expanding the Clinical and Genetic Spectrum of SPEG Variants: A Pediatric Case of Dilated-Left Ventricular Non-Compaction Cardiomyopathy with Compound Heterozygous Variants.

The present case is the first description of compound heterozygous SPEG mutations involving a de novo variant and causing dilated-LVNC without neuropathy or centronuclear myopathy.

 The data is exciting and could be of interest to the readers.

General comments:

  1. Either use mutation or variant. The term variant is preferred. Sequence Variant Nomenclature (hgvs.org). Kindly follow variant nomenclature.
  2. The article “The” is missing on many occasions in the manuscript.
  3. Kindly write the gene name in italic and the protein name in non-italic.
  4. Use a three-letter code for an amino acid.
  5. Add Sanger sequencing results of each member of the family.
  6. 3D protein modeling would be better to see the effect of the mutations on the secondary structures.
  7. Study approval was obtained? Mention the number.
  8. Frequency for SPEG:c.7408C>A, which is VUS is 0.000276, which is high. And the frequency for SPEG:c.8059C>A is also high (0.128) and classified as benign. How can the author justify the pathogenicity?

Title:

Very lengthy, can be reduced.

Abstract:

Two missense heterozygous mutations? Kindly write compound heterozygous variants.

Results

  1. Provide SD with growth parameters, e.g., height, weight, HC. If available.
  2. How many ethically matched controls was the variant screened to remove the occurrence of polymorphism?
  3. If abbreviated once, “Striated Muscle Enriched Protein Kinase (SPEG).” Next time only use SPEG.
  4. Father did not reveal any symptoms, although he had the same variant as the daughters.

Discussion

List all the variants reported so far along with their phenotypes, and see whether there are any genotype-phenotype correlations?

Materials and Methods

A flow chart explaining all the methodology will be helpful for the readers to understand in one sight.

Filtration steps used for variant prioritization?

Classification of the identified variant according to ACMG?
